# Effects of Molecular Crowding and Betaine on HSPB5 Interactions, with Target Proteins Differing in the Quaternary Structure and Aggregation Mechanism

**DOI:** 10.3390/ijms232315392

**Published:** 2022-12-06

**Authors:** Vera A. Borzova, Svetlana G. Roman, Anastasiya V. Pivovarova, Natalia A. Chebotareva

**Affiliations:** Bach Institute of Biochemistry, Federal Research Centre “Fundamentals of Biotechnology” of the Russian Academy of Sciences, Leninsky pr. 33, Moscow 119071, Russia

**Keywords:** HSPB5, chaperone-like activity, crowding, protein aggregation, betaine

## Abstract

The aggregation of intracellular proteins may be enhanced under stress. The expression of heat-shock proteins (HSPs) and the accumulation of osmolytes are among the cellular protective mechanisms in these conditions. In addition, one should remember that the cell environment is highly crowded. The antiaggregation activity of HSPB5 and the effect on it of either a crowding agent (polyethylene glycol (PEG)) or an osmolyte (betaine), or their mixture, were tested on the aggregation of two target proteins that differ in the order of aggregation with respect to the protein: thermal aggregation of glutamate dehydrogenase and DTT-induced aggregation of lysozyme. The kinetic analysis of the dynamic light-scattering data indicates that crowding can decrease the chaperone-like activity of HSPB5. Nonetheless, the analytical ultracentrifugation shows the protective effect of HSPB5, which retains protein aggregates in a soluble state. Overall, various additives may either improve or impair the antiaggregation activity of HSPB5 against different protein targets. The mixed crowding arising from the presence of PEG and 1 M betaine demonstrates an extraordinary effect on the oligomeric state of protein aggregates. The shift in the equilibrium of HSPB5 dynamic ensembles allows for the regulation of its antiaggregation activity. Crowding can modulate HSPB5 activity by affecting protein–protein interactions.

## 1. Introduction

HSPB5 (or αB-crystallin), a member of the small heat-shock protein (sHSP) superfamily, is an important component in the system of protein quality control in the cell [1]. sHSPs are ATP-independent chaperones, which bind unfolded protein molecules, thereby preventing their aggregation [1,2,3,4,5]. Under more severe stress conditions, sHSPs even tend to co-precipitate with their non-native substrates in aggregate-like assemblies [6]. Complexes of sHSPs with target proteins can be further utilized by proteasomal degradation pathways, or unfolded proteins can be refolded by ATP-dependent chaperones [6,7]. sHSPs are involved in many important processes in the cell, such as apoptosis, signal transduction, and other vital functions [4,6,7,8,9,10]. sHSPs appear to be engaged in the pathogenesis of disorders such as cancer [11], cataracts [12,13,14,15], and neurodegenerative diseases [5,7,15,16]. Therefore, ways of regulating sHSP functioning seems to be of great importance.

All sHSPs consist of an α-crystalline domain (ACD) and flexible C- and N-terminal extensions with an intrinsically disordered structures [17,18]. It is currently accepted that sHSPs form large, polydisperse multimeric assemblies through multiple weak interactions between N-/C-terminal extensions and ACD dimers [19]. Terminal domains are also involved in the interactions with unfolded target proteins [7,10,17,20,21]. All the domains are known to participate in the formation of heterooligomeric complexes [22,23]. Various oligomeric forms of sHSPs are established to display the chaperone-like (antiaggregation) activity [21,24,25]. Large multimers (consisting of up to 48–50 [20,26,27,28,29]) are currently believed to be the storage form [18,19]. When a target protein appears, these large oligomers undergo a rapid reorganization process, with partial dissociation into transient forms of smaller oligomers, dimers, and monomers [22,25]. These transient forms are thought to have higher antiaggregation activity and are capable of further dynamic reorganization [25].

The flexibility of the sHSP quaternary structure is currently being studied as one of the ways to regulate their functioning and activity [1,4,5,10,11,20,30,31]. It has been reported that sHSP dynamics have five levels of regulation [5], which include: (1) flexible domains flanking the ACD [17,18,19]; (2) polydisperse self-oligomerization; (3) heterooligomerization with other sHSPs [32,33]; (4) subunit exchange [7]; and (5) regulation by the cellular environment [5,8,10,11,25,31,34,35,36,37,38,39]. Crowding as a property of the intracellular environment is one of the levels of regulation [29,36,40,41,42,43].

It is implied that the cell is crowded with macromolecules (proteins, poly- and oligosaccharides, nucleic acids, etc. [44,45]) so that a part of the intracellular volume is unavailable for macromolecules [46]. This significantly influences the thermodynamic and kinetic parameters of their interaction, as compared with diluted buffer solutions in vitro [42,47,48,49,50,51,52].

Typically, the crowded milieu is mimicked in vitro by adding high concentrations of suitable inert polymers or proteins, so-called crowding agents, such as polyethylene glycols (PEG) of different molecular weights, polysaccharides (Ficoll and dextran), polyvinylpyrrolidone (PVP), or proteins (bovine serum albumin and lysozyme) [47]. It is currently believed that the effect of crowding can be considered as a mixture of the excludedvolume effect (EVE; an entropic factor) and soft interactions (an enthalpic factor) [51,53,54,55,56]. “Soft” interactions include electrostatic, hydrophobic, and van der Waals interactions between the crowding agent and a studied protein [57,58]. Both of these factors, entropic and enthalpic, can enhance as well as counteract the effects of each other [54,59,60,61,62,63,64].

Crowding, being the factor which markedly influences protein–protein interactions (including aggregation and supramolecular association) adds another degree of complexity to the dynamic structure–function relationships of sHSPs [20,29,36,40,41,42]. Our previous works [29,36,40,65,66,67,68] demonstrated the dramatic effect of crowding on the oligomeric state of several different sHSPs. It was also shown by our team in [43] that crowding significantly changed the antiaggregation activity of HSPB5 towards the heat-induced aggregation of glycogen phosphorylase *b* (Ph*b*).

Osmolytes (proline, betaine, TMAO, and others) are accumulated in the cell under stress conditions (osmotic stress, pressure changes, etc.). Osmolytes protect protein molecules from denaturation by stabilizing their tertiary structure [69]. Since the concentration of osmolytes in the cell can reach 1 M [70], they can contribute to the crowding effect [49]. It was reported that compatible osmolytes can also affect protein–protein interactions, such as association and aggregation [71,72]. It has been shown that the presence of a small crowder in a mixture containing macromolecules significantly increases the EVE [53,62], with the small crowder exhibiting a more prominent effect than macromolecular crowders [73]. It is believed that mixed crowding, created by model crowders of different chemical natures (charge and functional groups) and molecular weights, is a more reliable imitation of the intracellular environment [51,53,54,55,56,74]. It has been shown that a mixture of two crowders can influence the biochemical processes under study more effectively than a single macromolecular crowder at a high concentration [53].

It should be noted that the antiaggregation functioning of sHSPs is based on the interactions with target proteins that differ widely in size, mass, molecular structure, charge, and other parameters. It is reasonable to assume that the activity of sHSPs depends not only on their oligomeric structure and the environment, but also on the properties of the target protein. The attempt to quantify the activity of HSPB5 towards a target protein under crowding conditions was made in our previous work [43] with a heat-denatured Ph*b* used as the target protein. Crowding significantly reduced the chaperone-like activity of the HSPB5 in that case. Therefore, it is of interest to test how common this crowding effect is with other target proteins. The aim of the present work is to study the effects of a crowding agent (PEG), an osmolyte (1 M Bet), and their mixture on the chaperone-like activity of HSPB5. Bovine liver glutamate dehydrogenase (GDH) and hen egg white lysozyme (Lyz) were used as the target proteins. These proteins drastically differ in size, oligomeric structure, and aggregation kinetics.

Glutamate dehydrogenase (GDH) [EC 1.4.1.2-4] is one of the key metabolic enzymes which catalyze the oxidative deamination of L-glutamate to α-ketoglutarate [75]. The GDH from bovine liver used in this work is a homohexamer with a molecular weight of about 360 kDa [76]. GDH can form linear native associates in solution [75] and in vivo [77]. It makes GDH a rather complicated object for aggregation studies, as its supramolecular structure varies even under native conditions. GDH undergoes thermal denaturation with the formation of a molten globule-like intermediate, and this process is partially reversible up to 55 °C [78]. Earlier, we studied thermal aggregation of GDH at 50 °C [79] and showed that it proceeds as a reaction of the first order, with the rate-limiting stage being the protein unfolding. In this work, GDH was used as a target protein, with an oligomeric structure and a high molecular weight in contrast to a monomeric small globular protein lysozyme.

Lyz is a monomeric globular protein with *M*_w_ ~ 14 kDa, functioning as N-acetylmuramidase [80]. The Lyz molecule contains four disulfide bonds: one of them (between Cys6 and Cys127) is partially accessible to solvent; the other three are not accessible in the native protein [81]. However, they become exposed with DTT-induced protein denaturation, when the tertiary and secondary structures of Lyz are unfolded [82,83]. Lyz was shown to form amorphous aggregates under the reducing conditions at 37 °C and pH close to neutral [84]. Lyz has been used as a model protein to test the chaperone-like activity of HSPB5 [15,85]. In the present work, the kinetic regime of aggregation for the test system, based on the DTT-induced aggregation of Lyz, has been preliminary analyzed before considering the effects of the crowding agent (PEG), an osmolyte (1 M Bet), and their mixture on the chaperone-like activity of HSPB5.

## 2. Results

### 2.1. The Kinetics of DTT-Induced Aggregation of Lysozyme

The kinetic curves of DTT-induced aggregation of Lyz at 37 °C were registered by the increase in the light-scattering intensity (*I*) at several initial concentrations of the protein. As can be seen from Figure 1A, the aggregation intensifies with an increasing Lyz concentration. The initial parts of the curves corresponding to the nucleation stage were approximated by the following equation [43,86] (as an example, see curve 1, the inset in Figure 1A):(*I* − *I*_0_) = *K*_agg_ (*t* − *t*_0_)^2^, *t* > *t*_0_(1)
where *I*_0_ is the initial value of the light-scattering intensity at *t* = 0, *K*_agg_ is the parameter characterizing the acceleration of the aggregation at the nucleation stage (with the ongoing formation of nuclei), and *t*_0_ is the moment of time at which the initial increment of *I* is registered.

Parts of the kinetic curves after the inflection point, corresponding to the initial stage of the aggregate growth, were approximated by the following equation [87] (as an example, see curve 2, the inset in Figure 1A):(*I* − *I*_0_) = *v*_0_ (*t* − *t**) −*B*(*t* − *t**)^2^, *t* > *t**(2)
where *v*_0_ is the initial rate of aggregation at the aggregate growth stage, *t** is the duration of the lag period of this stage, and *B* is a constant.

As can be seen from panels B–D in Figure 1, the values of the parameters estimating the acceleration of the aggregation at the initial stage, *K*_agg_, and the aggregate growth rate at the later stage, *v*_0_, increase with increasing Lyz concentration, while the durations of the lag periods at this stage, *t*_0_ and *t**, monotonously decrease. To evaluate the order of aggregation with respect to the protein and the rate-limiting stage at the initial steps of the aggregation process, we analyzed the dependences of the kinetic parameters on the initial protein concentration ([Lyz]_0_). The equations that allow linking the parameters with the initial protein concentration ([P]_0_) are the following [43,86]:(3)$$K_{\mathrm{agg}}=const{\left[P\right]}_{0}^{b}$$
or
(4)$$v_{0}=const{\left[P\right]}_{0}^{c}$$
where *b* and *c* correspond to the order of aggregation with respect to the protein at the nucleation stage and the aggregate growth stage, respectively.

For DTT-induced aggregation of Lyz at 37 °C, *b* = 1.94 ± 0.13, as was determined using Equation (3) (Figure 1B). This means that the unfolding of Lyz molecules proceeds rather rapidly under the action of DTT, and the rate-limiting step at the initial stage of aggregation is the bimolecular reaction of aggregation of unfolded protein molecules. Thus, we may use the ratios of *K*_agg_^1/*b*^ ≡ *K*_agg_^1/2^ for estimating the effect of different agents on the nucleation stage of DTT-induced aggregation of Lyz [86].

The *c* value equals 0.96 ± 0.04, as was determined using Equation (4) (Figure 1D), i.e., the dependence of *v*_0_ on [Lyz]_0_ is linear. Thereby, the initial stage of the aggregate growth can be regarded as the reaction of the pseudo-first order, where the denatured monomers or small nuclei attach to the growing large aggregates with the constant concentration of the latter. The linear ratios of *v*_0_^1/c^ ≡ *v*_0_ may be used for estimating the effect of different agents on the initial aggregate growth stage of DTT-induced aggregation of Lyz.

### 2.2. The Effect of HSPB5, Betaine, PEG, and Their Mixture on Thermal Aggregation of GDH and DTT-Induced Aggregation of Lysozyme

The rate-limiting stage for GDH thermal aggregation at 50 °C was determined in our previous work [79]. The order of aggregation with respect to the protein, *n*, is equal to one. Therefore, the rate-limiting stage is the monomolecular process of GDH denaturation. Further aggregation proceeds through the relatively fast attachment of denatured molecules and dissociated forms of GDH to the growing aggregate. The power coefficients, *b* and *c*, from Equations (3) and (4) are also equal to one [79]. Thus, we may use the linear ratios of the parameters *K*_agg_ or *v*_0_ obtained in the absence and in the presence of different additives for estimating their effect on the initial stages of thermal aggregation of GDH.

The effect of HSPB5, 1 M Bet, 25 mg/mL PEG, and their mixture on the kinetics of the heat-induced aggregation of GDH at 50 °C or DTT-induced aggregation of Lyz at 37 °C was studied by the DLS method. Some series of the kinetic curves of aggregation of GDH and Lyz are represented in the Appendix A (Appendix A, respectively). As can be seen from these figures, HSPB5 holds both proteins from its aggregation in a concentration-dependent manner. It is conspicuous that the light-scattering intensity increases more slowly (for GDH) or at later times (for Lyz) in the presence of HSPB5. Analysis of all the obtained kinetic curves using Equations (1) and (2) allows us to obtain the quantitative expression of the antiaggregation effect of HSPB5. As it has been said before, the (*K*_agg_/*K*_agg,0_)^1/*b*^ and (*v*_0_/*v*_0,0_)^1/*c*^ ratios, where *K*_agg,0_ and *v*_0,0_ are the values of the parameters obtained in the absence of HSPB5, may be used for this purpose. The dependences of these ratios on the HSPB5 concentration are given in Figure 2.

Figure 2A shows that HSPB5 significantly suppresses the acceleration at the initial stage of GDH aggregation. The value of *K*_agg_/*K*_agg,0_ drops even at rather low concentrations of HSPB5 and tends asymptotically to zero at HSPB5 ≥ 0.075 mg/mL. The addition of 1 M Bet, 25 mg/mL PEG, or their mixture counteracts the antiaggregation activity of HSPB5 at this stage, although the effect in the presence of 1 M Bet is less than in the presence of PEG or the (PEG + 1 M Bet) mixture. Figure 2B demonstrates that HSPB5 also affects the stage of the aggregate growth. The value of the *v*_0_/*v*_0,0_ ratio decreases as the concentration of HSPB5 increases. The (*v*_0_/*v*_0,0_; [HSPB5]) dependence slightly changes in the presence of 1 M Bet or the (PEG + 1 M Bet) mixture. This indicates that HSPB5 retains its ability to influence this stage in the presence of these additives, albeit to a lesser extent. However, 25 mg/mL PEG favors a sufficiently active growth in the light-scattering intensity at this stage. As a result, the visible effect of HSPB5 reduces, as one may judge from the parameter *v*_0_ values.

In general, we observe a different extent of the decrease in the ability of HSPB5 to preserve GDH from aggregation at 50 °C in the presence of the crowding agent and/or the osmolyte.

As for the other test-system-DTT-induced aggregation of Lyz at 37 °C—the results of probing the antiaggregation activity of HSPB5 in the presence of the crowding agent and/or the osmolyte are presented in Figure 2, panels C and D. Figure 2C shows that the values of the (*K*_agg_/*K*_agg,0_)^1/2^ ratio increase by up to 1.8 times for the aggregation of Lyz in the presence of 0.001–0.0075 mg/mL HSPB5. Therefore, an acceleration of Lyz aggregation takes place at relatively low concentrations of HSPB5. At concentrations higher than 0.01 mg/mL, HSPB5 suppresses the initial stage of DTT-induced Lyz aggregation. In the presence of 1 M Bet, 25 mg/mL PEG, or their mixture, HSPB5 suppresses the initial stage of Lyz aggregation even better. All the points corresponding to the (*K*_agg_/*K*_agg,0_)^1/2^ ratio at various HSPB5 concentrations obtained in the presence of the additives lie under those measured just in the buffer. Moreover, there is no more acceleration of Lyz aggregation observed at low concentrations of HSPB5.

Figure 2D depicts the dependences of the *v*_0_/*v*_0,0_ ratios on the HSPB5 concentration. As can be seen from the figure, the initial aggregation rate at the aggregate growth stage increases in the presence of low concentrations of HSPB5 and apparently decreases at HSPB5 > 0.01 mg/mL. The presence of PEG in the buffer leads to a reduction in the *v*_0_/*v*_0,0_ value at concentrations of HSPB5 up to 0.01 mg/mL, whereas the effect of higher concentrations of HSPB5 does not change. The addition of 1 M Bet and its mixture with PEG alters the dependence of the *v*_0_/*v*_0,0_ ratio on HSPB5. The *v*_0_/*v*_0,0_ ratio increases in the presence of the osmolyte compared to that measured in the absence of any additives and does not decrease with rising concentrations of HSPB5. When both the osmolyte and the crowding agent are present in the mixture, the *v*_0_/*v*_0,0_ values are even higher—by up to 2.5 times—than those obtained just in the buffer.

Thus, the antiaggregation activity of HSPB5 enhances in the presence of the crowding agent and/or the osmolyte at the nucleation stage (judging from the values of the (*K*_agg_/*K*_agg,0_)^1/2^ ratios). At the same time, the presence of the osmolyte, as well as its mixture with the crowding agent, may lead to more rapid aggregation at the aggregate growth stage.

Considering the above data on the aggregation of both GDH and Lyz, one could speak about a possible negative effect of the osmolyte on the antiaggregation activity of the sHSPs and an expectation of aggregation process aggravation under these conditions. However, if we compare the absolute (not relative) values of the kinetic parameters measured in the presence of HSPB5 in dilute or under crowded conditions (arising from the presence of PEG) with those measured in the presence of HSPB5 and the osmolyte, we will see a significant suppression of the aggregation of GDH as well as Lyz in the presence of the osmolyte, even under crowded conditions. The diagram in Figure 3 clearly demonstrates this for the case of HSPB5 = 0.05 mg/mL. It can be seen from the diagram that, while the crowding agent (25 mg/mL PEG) may have a different effect on the aggregation of the target protein, the osmolyte (1 M Bet) always suppresses it (Figure 3). In the presence of HSPB5, this suppression remains essential, despite the fact that it may be reduced in the presence of the osmolyte with/without the crowding agent (cf. the shaded and gray “betaine”/“PEG+betaine”-labeled bars in Figure 3, panels A, B, and D).

### 2.3. Sedimentation Velocity Analysis of HSPB5 Interaction with Lysozyme in the Presence of Betaine, PEG, and Their Mixture

HSPB5 and other agents were added to Lyz (denatured by 20 mM DTT during 15 min at 37 °C, cooled, and then dialyzed overnight), followed by AUC experiments at 37 °C. As has been previously said, the rate-limiting step at the initial stages of DTT-induced aggregation of Lyz is the aggregation of denatured protein molecules (see Section 2.1). This fact validates the sedimentation experiments when HSPB5 and other agents are added to previously denatured proteins.

The differential distribution, *c*(*s*), for Lyz reveals two major peaks, with sedimentation coefficients of 1.1 ± 0.4 and 2.1 ± 0.4 S (Figure 4), which indicate the presence of monomeric and dimeric forms of the protein in the solution. In addition, there are minor peaks (at 3.2 S and more) corresponding to small aggregates. The addition of HSPB5 at concentrations of 0.01, 0.05, or 0.15 mg/mL did not strongly affect the position of the main peaks in the distribution. Figure 4 shows that the presence of HSPB5 at the highest concentration used, 0.15 mg/mL, slightly increases the *s*_20,w_ value for small aggregates from 3.2 to 3.6 S. It may indicate the formation of complexes between the monomeric or dimeric HSPB5 forms with the Lyz monomer or dimer. The appearance of broad peaks at 11.4, 14, and 16 S indicates the presence of free HSPB5 associates or their high-molecular-weight complexes with denatured Lyz.

The effect of the osmolyte (1 M Bet), the crowding agent (25 mg/mL PEG), or their mixture (1 M Bet + 25 mg/mL PEG) on the interaction of HSPB5 (0.05 mg/mL) with Lyz is shown in Figure 5.

In the presence of 1 M Bet and HSPB5, the predominantly monomeric form of lysozyme with the sedimentation coefficient *s*_20,w_ of 1.5 S is stabilized (red dash-dotted curve in Figure 5). In addition, small aggregates (3.4, 5.3, 8.9, 10 S, etc.) are formed. If crowding conditions are created only by the high-molecular-weight polymer PEG (25 mg/mL), then the *c*(*s*) distribution for the mixture of lysozyme with HSPB5 reveals the main peak at 3.3 S (blue dashed line in Figure 5) (with estimated mass 32.7 kDa) and a shoulder at 4.6 S (with estimated mass 53.5 kDa). This indicates the presence of both the small Lyz aggregates and the possible complexes of the Lyz monomer with the chaperone monomer or dimer. The presence of a minor peak at 1.2 S indicates the presence of a partially unfolded monomeric form of lysozyme.

Under the mixedcrowding conditions arising from the presence of 1 M Bet and 25 mg/mL PEG, the *c*(*s*) distribution shifts towards higher sedimentation coefficients (green solid line in Figure 5). The broad peak at 3.2 ± 0.4 S indicates the presence of small lysozyme aggregates, as well as possible complexes of the Lyz monomers and dimers with the monomeric or dimeric forms of HSPB5. The 6.5 S shoulder on the distribution shows the presence of small aggregates of lysozyme, which may contain incorporated HSPB5 molecules. In addition, there are large aggregates with *s*_20,w_ = 13.3 and 26 S (not shown), which indicate the presence of free HSPB5 or its high-molecular-weight complexes with denatured Lyz.

### 2.4. Sedimentation Velocity Analysis of the Interaction of HSPB5 with GDH in the Presence of Betaine, PEG, and Their Mixture

After 10 min of heating GDH at 50 °C, a significant part of the protein precipitates during the acceleration of the rotor (58%, Table 1) in SV experiments. The addition of HSPB5 (0.05 mg/mL) to the sample before the heating partially stabilizes the hexameric form of GDH (peak at 14.9 S) and small aggregates or complexes with the sHSP (the shoulder at 16 S and the peak at 39 S), as can be seen in Figure 6. The fraction of precipitated protein γ_agg_ was 57% (Table 1).

The presence of HSPB5 and the crowding agent PEG preserves GDH from precipitating and leads to a shift of the *c*(*s*) distribution towards higher values of sedimentation coefficients, which correspond to larger particles. The *c*(*s*) distribution becomes broad, with two peaks at 16 and 19 S. This shows that crowding created by PEG can stabilize the hexameric form of GDH, its small oligomers, and their complexes with suboligomeric forms of HSPB5 (γ_agg_ = 0, Table 1).

In the presence of the natural osmolyte, 1 M Bet, the protein does not precipitate (Table 1) and the entire *c*(*s*) distribution shifts towards lower sedimentation coefficients with an average value of *s*_20,w_ = 11.9 ± 1.1 S. These data suggest that the dissociated forms of GDH and their complexes with HSPB5 are stabilized in the presence of 1 M Bet.

Under mixed crowding conditions arising from the presence of a mixture of 1 M Bet + PEG, the entire distribution *c*(*s*) becomes very polydisperse and shifts towards high-molecular-weight particles with *s*_20,w_ = 31.7 ± 1.8 S and a minor peak at 14.7 S corresponding to the GDH hexamer (Figure 6). A broad peak in the distribution indicates the presence of large complexes, with a mass ranging from 830 kDa to several MDa. High-molecular-weight particles can be both aggregates and complexes of GDH with HSPB5. We can speak about the formation of complexes, since, due to the low concentration and small extinction coefficient, the contribution of HSPB5 to the optical absorption of the sample is insignificant. However, comparison of the *c*(*s*) distribution for GDH (0.2 mg/mL) under mixedcrowding conditions (PEG + 1 M Bet) with the *c*(*s*) distribution of a mixture of GDH and HSPB5 (0.05 mg/mL) (inset in Figure 6) shows a strong distribution shift towards lower sedimentation coefficients in the presence of the sHSP. In addition, Table 1 shows that, in the presence of PEG and Bet, 50% of GDH precipitates, and the addition of HSPB5 protects the protein from precipitation (γ_agg_ = 0) under mixedcrowding conditions. Thus, despite the fact that the presence of the osmolyte or PEG affects the sedimentation of a mixture of GDH with HSPB5 (0.05 mg/mL) in the opposite way—namely, osmolyte stabilizes smaller oligomeric forms of GDH, and PEG stabilizes large GDH associates—mixed crowding dramatically shifts the distribution towards high-molecular-weight associates/aggregates.

## 3. Discussion

The dynamic quaternary structure of sHSPs, particularly the subunit exchange and the subsequent reorganization of oligomers, is an important factor in the regulation of their antiaggregation activity [6,7,11,25,31,88,89]. Small oligomeric forms (monomers and dimers) are thought to be the major units of the exchange between the multimeric sHSP assemblies. It was suggested [89] that the content of monomers and dimers in oligomers and their exchange can finetune the functioning of sHSPs since the assemblies with monomeric and dimeric building blocks differ in the conformation and the antiaggregation activity. Monomers have more exposed hydrophobic areas and are considered as the most active species preventing the aggregation of target proteins, even at low concentrations [25,89,90]. However, an exposed hydrophobic surface is unfavorable in terms of solubility [91]. It has been suggested that the coexistence of traveling subunits and normal oligomers in αB-crystallin is biologically necessary for balancing chaperone activity and proper solubility [31]. Crowding can modulate these processes by affecting protein–protein interactions. 

In the present work, the effects of crowding and an osmolyte, as well as their combined effect on the antiaggregation activity of HSPB5, have been investigated. It is important to note that the activity of sHSPs also depends on the target protein. Two test-systems with different kinetic regimes of the protein aggregation were used. One of them is based on the thermal aggregation of GDH and other on the DTT-induced aggregation of Lyz at 37 °C. To quantify and compare the effects of the osmolyte and crowding on the antiaggregation activity of HSPB5, we calculated the parameters *K*_agg_ (the acceleration of the aggregation at the nucleation stage) and *v*_0_ (the initial rate at the stage of the aggregate growth) from the kinetic curves obtained by DLS.

GDH is a hexameric protein with *M*_w_ ~ 360 kDa. The denaturation of the protein was established to be the rate-limiting step for GDH aggregation at 50 °C [79]. Thus, the formation of the intermediate states and the protein unfolding during the heat-induced transition of GDH from the native to the denatured state proceeds at a much lesser rate than the subsequent aggregation of the denatured GDH molecules.

As a starting point, we considered the effects of Bet and crowding on the chaperone-like activity of HSPB5, using GDH as a target protein. In the absence of all the additives, HSPB5 reduces GDH aggregation in a concentration-dependent manner (Figure 2 and Appendix A), although it is not capable of protecting the system from the formation of high-molecular-weight insoluble aggregates. The AUC data show that 0.05 mg/mL HSPB5 does not diminish the fraction of the large aggregates precipitating during the rotor acceleration in the SV runs (γ_agg_ = 57 versus 58%, Table 1). When placed in the more “cell-like” conditions containing the macromolecular crowder (PEG), the osmolyte (Bet), or their mixture, the antiaggregation activity of HSPB5 decreases, judging from the aggregation kinetics registered by the DLS method (Figure 3A,C), i.e., HSPB5 would suppress the process much more if there were no additives. It is known that crowding (as the excluded volume effect) may result in the stabilization of more compact protein states and thereby impede the complete unfolding of the protein molecules during denaturation. The effect of PEG, Bet, and their mixture on GDH thermal aggregation kinetics seems to correspond to this case, especially since denaturation is the rate-limiting stage for the process. It should be noted that Bet, as well as other compatible osmolytes, is often considered as a chemical chaperone—the low molecular compound preventing protein aggregation. The simple explanation for the relative decrease in the chaperone-like activity of HSPB5 could be that the stabilization of the target protein precludes the interaction of the chaperone with it. However, we should not ignore the influence of Bet, PEG, and their mixture on the structure and dynamics of HSPB5 and possible effects of that on the interaction with the target protein.

Remarkably, according to the AUC data, HSPB5 reveals an antiaggregation activity that is sufficient to hold the aggregates in a soluble state for quite a long time (γ_agg_ = 0, Table 1) in the presence of PEG, 1 M Bet, or their mixture.

The comparison of the differential sedimentation coefficient distributions, *c*(*s*), for the pair of GDH and HSPB5 in the absence and in the presence of PEG, or 1 M Bet, or their mixture (in Figure 6), shows the most prominent effect of HSPB5 on GDH aggregation in the presence of the mixture of Bet and PEG. The AUC results demonstrate that 1 M Bet, with and without HSPB5, prevents the precipitation of GDH (Table 1) and stabilizes its dissociated non-native oligomeric forms with the sedimentation coefficients less than that of the native GDH hexamer (Figure 6 and Appendix A, the *s*_20,w_ value for the native GDH hexamer is 14.7 S [79]). We have shown earlier that other chemical chaperones (proline, arginine, and its derivatives) can prevent thermal aggregation of GDH by stabilizing dissociated protein forms [79].

Crowding arising from the presence of PEG decreases the aggregation of GDH (Figure 3 and Appendix A), increases the size of aggregates (Appendix A) (possibly due to the EVE), and partially protects GDH from precipitation. About 16% of the aggregated GDH precipitate in the presence of the crowding agent versus 58% in the dilute buffer (Table 1). In the presence of HSPB5, the target protein is completely protected from precipitation (γ_agg_ = 0, Table 1). The *c*(*s*) distributions demonstrate the preservation of a range of soluble GDH aggregates (16–26 S, Appendix A) in the presence of PEG, while smaller aggregates (up to dodecamers) or possible complexes of different forms of GDH with different HSPB5 suboligomers are stabilized in the presence of HSPB5 (16–19 S, Figure 6).

The presence of a mixture of Bet and PEG in the system results in the formation of larger and more polydisperse protein complexes, both in the absence of HSPB5 (Appendix A) and in the presence of the chaperone (Figure 6). This unexpected/nonadditive effect could be explained by mixed crowding. Bet, at a fairly high concentration (1 M), can be considered as a crowding agent [49]. It was reported that two crowders can exhibit a synergistic effect, significantly enhancing the effect of each other, even when using relatively small concentrations (10–20 mg/mL) [43,53]. It has also been shown that small crowders create more total excluded volume in the vicinity of a big crowder than in the bulk [53,62]. To study the effects of size polydispersity of crowders on the aggregation reaction equilibrium, Shah and his colleagues developed a molecular thermodynamic formalism. They showed that, in the case of polydisperse crowders, the crowders with the largest size differences dominate the overall changes in reaction yield [54]. In our case, it can result in the enhancement of protein aggregation and association.

The comparison of the *c*(*s*) distributions, given in the inset in Figure 6, shows the prominent effect of HSPB5 on GDH in the condition of mixed crowding. It is even more evident when comparing γ_agg_ values from Table 1 (50% in the absence of the chaperone and 0% in the presence of HSPB5). Therefore, the addition of HSPB5 to the aggregating GDH under the mixedcrowding condition results, apparently, in the effective interaction of the chaperone with the target protein. In addition, the protection of GDH from precipitation in the presence of HSPB5 could not be attributed to the solution viscosity; although it is relatively high for the mixture of Bet and PEG (Appendix A), it does not change in the presence of the chaperone.

It is worth noting that during the thermal aggregation of Ph*b* (a dimeric protein with *M*_w_ of 195 kDa), at 48 °C the antiaggregation activity of HSPB5 decreased much more significantly under crowding conditions—up to 55 times (as evaluated by the parameter *K*_agg_) [43]. However, small complexes formed by suboligomeric forms of HSPB5 (monomers or dimers) with Ph*b* monomers/dimers and small oligomers remained in solution for a long time (90 min) according to SV data [43]. Here, we may also state the interaction between HSPB5 suboligomers and GDH trimers/tetramers (in the presence of 1 M Bet) or hexamers and their small associates (in the presence of PEG).

It should also be noted that the aggregation of target proteins in the presence of sHSPs, even with the apparent decrease in antiaggregation activity of the chaperone, could be a part of overall protein quality control in the cell. It was shown [6,18,92] that aggregates formed in the presence of sHSPs contain incorporated subunits of chaperones, which facilitate further disaggregation by ATP-dependent chaperones, followed by possible refolding of the target protein and recycling of sHSPs. These chaperone-containing aggregates are shown to be smaller and more compact than the amorphous aggregates without sHSPs [92]. We estimated by DLS the hydrodynamic radius (*R*_h_) of GDH aggregates after 10 min of heat-induced aggregation at 50 °C, in the absence and in the presence of HSPB5. It was found (Appendix A) that in the absence of the chaperone, the mean *R*_h_ of aggregates was 782 nm, whereas in the presence of 0.05 mg/mL of HSPB5, the mean *R*_h_ was 136 nm. Therefore, it was confirmed that aggregates of smaller size were formed in the presence of the chaperone.

Some other conclusions might be drawn when another test-system based on DTT-induced aggregation of Lyz at 37 °C is used. Unlike GDH, lysozyme is a relatively small, compact protein with a molecular mass of 14 kDa, less than that of the HSPB5 subunit (*M*_w_ ~ 20 kDa). Another difference is in the kinetic regime of aggregation. Under the conditions studied (20 mM DTT, 37 °C), the denaturation of Lyz is fast and the rate-limiting stage of the overall aggregation process is the stage of aggregation of unfolded protein molecules (see Section 2.1).

It is interesting that, in the case of DTT-induced Lyz aggregation, relatively low concentrations of HPSB5 (<0.01 mg/mL) significantly accelerate the nucleation stage (Figure 2C) and increase the rate of the aggregate growth stage, although to a lesser extent (Figure 2D). The similar results were obtained earlier for α-crystallin from a bovine eye lens and DTT-induced aggregation of human recombinant insulin [93]. The authors suggested that α-crystallin particles could act as aggregation seeds at low concentrations of the sHSP, accelerating the aggregation of the target protein. Recently, it has been shown that the high affinity of the HSP27 monomeric form to a porcine heart citrate synthase during thermal aggregation of the latter is likely to result in the coaggregation of the HSP27-denatured protein complexes, promoting the aggregation [34]. Considering these data, the enhancement of Lyz aggregation by low concentrations of HSPB5 observed in the current work may be explained in the context of the dynamic structure of sHSPs. If the concentration of sHSPs is low enough, it can be suggested that the assembly/disassembly equilibrium among the forms of sHSPs would be shifted to the suboligomeric forms. If the solubility of the latter is lower, the irreversible coaggregation of the sHSP–target protein may occur. In this case, sHSPs could indeed act as an aggregation promotor at low concentrations. It was described earlier [18] that a low ratio of sHSPs/substrate could lead to the formation of insoluble protein–protein complexes, increasing the solution turbidity, whereas at the high sHSPs/substrate ratios, the complexes formed are soluble. It should also be remembered here that even large coprecipitated complexes of sHSPs with the target protein could undergo disaggregation and refolding in vivo.

The addition of the osmolyte and/or the crowding agent completely cancels this effect of acceleration at the nucleation stage (Figure 2C), resulting in the suppression of aggregation even by low concentrations of HSPB5. This may be important in the crowded environment of a cell. For example, at the initial times of stress reaction, when the expression of sHSPs is not yet enhanced, it could be necessary to suppress the nucleation process and the detrimental promotion of aggregation by insufficiently high concentrations of the chaperone at the initial stage of aggregation.

The aggregation-promoting effect of relatively low concentrations of HSPB5 also manifests itself at the stage of the aggregate growth. This effect of HSPB5 is even higher in the presence of Bet and its mixture with PEG. Both Bet and the mixture of additives diminish the protective action of any studied HSPB5 concentrations at this stage (Figure 2D and Figure 3D). On the contrary, the aggregation suppression by relatively low concentrations of HSPB5 (below 0.025 mg/mL) is better in the presence of PEG alone (Figure 2D and Figure 3D). The interaction of Bet with Lyz aggregates and HSPB5 is supposed to have an impact on the acceleration of aggregation at this late stage of the process, whereas PEG continues to improve the chaperone-like activity of HSPB5, such as during the nucleation stage (Figure 3C,D). What is important here is that the presence of a mixture of PEG with Bet results in a nonadditive effect on the rate of the aggregate growth (Figure 3D). The AUC data also show that the *c*(*s*) distribution is shifted towards larger sedimentation coefficients (larger-sized particles) in the presence of both Bet and PEG, in contrast with the individual effects of PEG and Bet (Figure 5 and Appendix A). Therefore, the mixed crowding affects this aggregation test system similarly to the previously described thermal aggregation of GDH. It can be speculated that the extent of this influence can depend on the size of the target protein.

The complicated and sometimes counteracting effects of the crowder or/and the osmolyte on the different stages of the protein aggregation process could be explained by the interaction of the additives with the target protein, including soft interactions. The effect of individual agents and their mixture on target proteins without a chaperone is demonstrated in Appendix A. It is useful to remind here that various soft interactions are essential for the functioning of a protein in a crowded environment, although molecular crowding is often treated as an excluded volume effect. Even when protein molecules are considered as hard particles in a crowded media, it was shown that the resulting effect on the protein–protein interaction depends on the shape and the structure of the target protein [63]. This work also considers the fact that different types of crowding agents (including high concentrations of osmolytes and protein denaturants, sugars, synthetic polymers, and other proteins) can dramatically differ in their effects on the same target, up to the direct contradiction to theoretical expectations. In the current work, we use Bet, which is a natural osmolyte that can not only stabilize proteins under stress but also modulate their structural dynamics via direct interaction with the protein molecule [69]. This can contribute to the observed intricate effects on the protein aggregation. It should be understood that the aggregating test system is a multicomponent system of a protein in a native state and in various forms of denatured and dissociated intermediates, as well as a polydisperse set of aggregates, all of these differing in size, shape/compactness, and surface properties. HSPB5, another component of the studied systems, is also a polydisperse protein that dynamically responds to the presence of the above factors (osmolyte, crowders, and target proteins) with structural alterations. Moreover, HSPB5 is known to interact with different targets by different regions of its molecule. For example, Lyz is bound at the N-terminal region of HSPB5 [94]. The substrate may define the binding region in the sHSP molecule, whether it is the core domain or the terminal regions. This may affect the quaternary dynamics of sHSPs and the stoichiometry of the complexes with target proteins. We cannot speculate about any substrate preference of HSPB5 based on our data, especially knowing that HSPB5 is a chaperone with a broad target spectrum [95]. However, the difference in the kinetic regime of aggregation of the target proteins is likely to be important. In the case of a slow, rate-limiting denaturation of GDH, the chaperone can capture the intermediate forms of GDH at the early stage of its unfolding. It was described in the literature [18] that sHSPs can recognize and bind substrates at the early stage of unfolding and preserve most of their native conformation. In our case, even relatively low concentrations of HSPB5 are able to inhibit the formation of large aggregates of slowly denaturing GDH, whereas there is an ambiguous effect of the chaperone on the aggregation of a rapidly denaturing small lysozyme.

Our work attempts to characterize systems with such a complex and constantly changing structure from a functional point of view by quantifying the chaperone-like activity of HSPB5. We used kinetic analysis and directly observed the outcome of the interaction of the chaperone with the targets using AUC. This functional approach also has physiological relevance. It can be assumed that, in vivo and under stress conditions, the accumulation of osmolytes and the formation of cytoplasmic chaperone-containing foci [96] with a possible increase in macromolecular crowding, may contribute to the effectiveness of molecular chaperones such as sHSPs. An increase in the concentration of HSPB5 and its activity under stress play a protective role in the cell, but there are situations when the activity of the chaperone should be reduced, for example, in cancer cells or in the case of pathological protein deposition (cataract and cardiomyopathy) [97]. Crowding should be considered as one of the factors that may subtly modulate HSPB5 activity. It affects the quaternary structure of HSPB5, the subunit exchange, the conformation of subunits (especially the intrinsically disordered N- and C-terminal extensions), and the interaction with substrates.

## 4. Materials and Methods

### 4.1. Materials

Bovine liver GDH (suspension in ammonium sulfate), hen egg white lysozyme, mono- and dibasic sodium phosphates, sodium chloride, and polyethylene glycol with a molecular mass of 20,000 Da (PEG) were purchased from Sigma-Aldrich (St. Louis, MO, USA). 1,4-dithiothreitol (DTT) was purchased from PanReac (Barcelona, Spain). Betaine was purchased from ICN Biomedicals (Costa Mesa, CA, USA). All solutions for the experiments were prepared using deionized water obtained with the Arium Mini system (Sartorius, Göttingen, Germany) and Simplicity UV system (Millipore, Burlington, MA, USA). 

All samples and stock solutions were prepared in 0.1 M Na-phosphate buffer, pH 7.6, and 10 mM NaCl.

### 4.2. Protein Preparations

GDH was purified from the suspension using PD-10 desalting column (GE Healthcare, Little Chalfont, Buckinghamshire, UK) according to the instructions. The human HSPB5 coding sequence was cloned into the pET23 vector for expression in *E. coli* cells, as described in [98]. Overexpressed HSPB5 was purified by salting out with ammonium sulfate, followed by the gel filtration. The purest fractions (according to SDS gel electrophoresis) were combined, concentrated, aliquoted, and stored at −80 °C.

Lyz solutions were prepared by dissolving the lyophilized protein in the buffer, with following centrifugation at 20,000× *g* and +4 °C for 30 min in the Eppendorf 5417R centrifuge to remove insoluble particles. HSPB5 preparation was dialyzed overnight at +4 °C against the working buffer (0.1 M Na-phosphate, pH 7.6, and 10 mM NaCl) prior to the experiments. Protein concentrations were determined using Spekol 1300 (Analytik Jena, Jena, Germany) spectrophotometer. The extinction coefficients of 1 mg/mL protein, *ε*_0.1%_^280^, were equal to 0.97 for GDH [99], 2.69 for Lyz [100], and 0.693 for HSPB5 [98].

### 4.3. Dynamic Light Scattering (DLS)

Aggregation kinetics was registered by an increase in the light-scattering intensity using the Photocor Complex correlation spectrophotometer (PhotoCor Instruments, Inc. College Park, MD, USA) with a He-Ne laser (Model 31-2082, 632.8 nm, 10 mW, Coherent Inc., Santa Clara, CA, USA) as a light source. The scattered light was collected at a 90° angle, and the accumulation time of the autocorrelation functions was 30 s for each experimental point. Glass vials with stoppers were preheated at the desired temperature (50 °C for GDH and 37 °C for Lyz) with the buffer and with/without the crowding agent and/or the osmolyte. Then, the small aliquot of HSPB5 was added, if necessary. In the experiments with Lyz, the protein was also preheated in the vial and the aggregation process was initiated by the addition of DTT to the final concentration of 20 mM. In the experiments with GDH, the aggregation process was initiated by adding the protein into the preheated solution. All experiments were repeated in triplicate.

### 4.4. Analytical Ultracentrifugation (AUC)

Sedimentation velocity (SV) experiments were carried out using a Model E analytical ultracentrifuge (Beckman Instruments, Palo Alto, CA, USA) with absorbance optics, a photoelectric scanner, a monochromator, and an online computer. A four-hole rotor An-F Ti and 12 mm double-sector cells were used. The rotor was preheated in a thermostat at either 37 °C or 25 °C overnight before the run.

In the case of Lyz, SV runs were performed at 37 °C in 0.1 M Na-phosphate buffer, pH 7.6, containing 10 mM NaCl, with or without the crowding agent and/or the osmolyte. The rotor speed was 60,000 rpm or 48,000 rpm. The solution of Lyz ([Lyz] = 1 mg/mL) was incubated for 15 min at 37 °C in the presence of 20 mM DTT, centrifuged 30 min at 20,000× *g* and 4 °C to remove aggregated particles and prevent complete precipitation, and then dialyzed overnight at 4 °C against the buffer without DTT for the AUC experiments. This was done to remove DTT, which interferes with the optical absorbance signal in the sedimentation experiments. After that, HSPB5 and/or the crowding agent and/or the osmolyte were added, if necessary, immediately before the run.

In the case of GDH, SV runs were performed at 25 °C in 0.1 M Na-phosphate buffer, pH 7.6, containing 0.01 M NaCl, with or without the crowding agent, or the osmolyte, or the mixture of both. The rotor speed was 48,000 rpm. All GDH samples were incubated for 10 min at 50 °C before the run.

Sedimentation profiles were recorded by measuring the absorbance at 280 nm. All cells were scanned simultaneously. The time interval between scans was 2.5 min. The differential sedimentation coefficient distributions (*c*(*s*) versus *s*) were determined using SEDFIT program [101]. Sedimentation coefficients were corrected to the standard conditions (a solvent with the density and viscosity of water at 20 °C) using SEDFIT [101]. The values of the density and the dynamic viscosity of the solutions used in the AUC measurements at 25 °C and 37 °C are presented in Appendix A of the Appendix A.

### 4.5. Densitometry and Viscosimetry

The dynamic viscosities (η) of the buffer, Bet, and PEG solutions, and their mixture, were measured at 25 and 37 °C using an AMVn microviscometer (Anton Paar, Graz, Austria) with a 1.6/1.500 capillary system. The densities (ρ) of these solutions were measured at 37 and 50 °C with the DMA 4500 densitometer (Anton Paar, Graz, Austria). The obtained values of densities and viscosities are given in the Appendix A.

## 5. Conclusions

Crowding adds another level of complexity not only to the regulation of sHSPs function, but to the protein aggregation as well. The net effect of crowding on protein–protein interactions depends on the shape and other physical parameters of the target proteins, and on the sum of interactions of different types of crowders with these proteins. Some of the compounds that can create the crowding effect also have other ways to influence protein structure and stability. For example, Bet is an osmolyte and may be considered as a crowder (at high concentrations). It should also be remembered that HSPB5 assemblies are highly dynamical structures, and changes in their composition are linked to the regulation of substrate specificity and activity of HSPB5 [6]. The system, consisting of proteins of high polydispersity (i.e., aggregating protein with its full range of intermediates), HSPB5 multimers, and small oligomers, which can also dynamically change in response to substrate and environment, is an incredibly complex object in terms of protein–protein interactions.

In this work, we approached the analysis of such a system from the side of a protein function, which is a direct result of all the above-mentioned complicated and dynamic interactions. Our point was to characterize and compare this protein function, namely the chaperone-like activity of HSPB5, at the different stages of protein aggregation using target proteins with different sizes, oligomeric structures, and aggregation kinetic regimes. To quantify the chaperone-like activity of HSPB5, an analysis was conducted of the aggregation kinetic curves obtained by DLS. The oligomeric state of the target proteins and their aggregates upon the interaction with HSPB5, in the presence and in the absence of osmolyte and/or crowder, was monitored by AUC.

It was shown that, in the case of heat-induced aggregation of GDH at 50 °C (the rate-limiting stage is the stage of protein denaturation), HSPB5 suppresses aggregation at both nucleation and subsequent aggregate growth stages. In the presence of 1 M Bet, 25 mg/mL PEG, and their mixture, the activity of HSPB5 decreased, judging by the kinetic parameters (the relative acceleration at the nucleation stage and the relative initial rate of aggregate growth; the relative term refers to the comparison of the parameter value in the presence and in the absence of the chaperone). However, the AUC data showed that, although in some cases crowding and osmolytes caused the formation of larger aggregates, the presence of HSPB5 resulted in the prevention of GDH precipitation. The protein aggregates remained soluble. Thus, even the decreased activity of HSPB5 was sufficient in the long-term protection of the target protein.

In contrast, the DTT-induced aggregation of Lyz at 37 °C has the rate-limiting stage of the aggregation of unfolded protein molecules (the denaturation proceeds rapidly). We observed mixed and often controversial effects at different stages of aggregation in the presence of different additives and concentrations of HSPB5. While the previous test system with the rate-limiting stage of denaturation primarily reflected in its aggregation kinetics the changes in the target protein stability, the Lyz-based system revealed all the complexity of possible ways of interaction between denatured protein, HSPB5, osmolyte, and polymeric crowder. We observed the acceleration of aggregation by low concentrations of HSPB5 (0.01–0.025 mg/mL), and the prevention as well as promotion of this effect by Bet and PEG at different aggregation stages. The higher concentrations of HSPB5 suppressed Lyz aggregation and Bet, and PEG caused an increase in chaperone-like activity in some cases, and a decrease in other cases. Nevertheless, the AUC data also showed the overall protective effect of high enough concentrations of HSPB5 (0.05 mg/mL) on the partially denatured target protein at 37 °C. The general effect of the chaperone in the presence of Bet, PEG, and their mixture on the comparative size/mass of protein aggregates was mostly a decrease in the aggregates size and the changes in the conformation of Lyz monomers and dimers.

This work provides information for a deeper understanding of the complex relationship between structure and function of sHSPs in more native-like surroundings than the usually studied dilute buffer solutions. It may be useful for further research in the field of molecular chaperones, their functioning, regulation, and involvement in physiological and pathological processes, and may provide possible applications in the treatment of the latter.

## Figures and Tables

**Figure 1 ijms-23-15392-f001:**
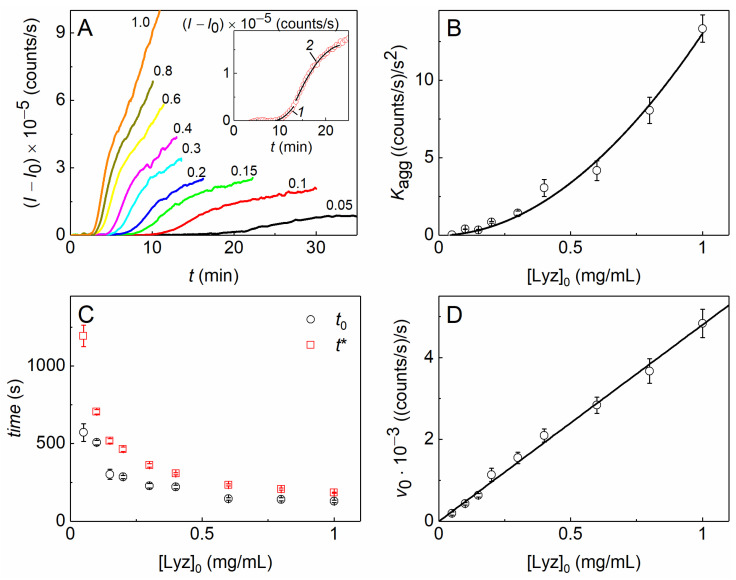
The kinetics of DTT-induced aggregation of Lyz (0.1 M Na-phosphate buffer, pH 7.6, 10 mM NaCl, 20 mM DTT) at 37 °C. (**A**) The growth of the light-scattering intensity from its initial value (*I* − *I*_0_) over the time (*t*) at different initial concentrations of Lyz (numbers near the curves correspond to Lyz concentration in mg/mL). The inset shows the kinetic curve for 0.1 mg/mL Lyz as an example of calculating the kinetic parameters: curves 1 and 2 are calculated using Equations (1) and (2), respectively. (**B**) The dependence of *K*_agg_ on the initial concentration of Lyz. The solid curve was calculated using Equation (3), with *b* = 1.94 ± 0.13. (**C**) The dependence of *t*_0_ and *t** on the initial concentration of Lyz. (**D**) The dependence of *v*_0_ on the initial concentration of Lyz. The solid curve was calculated using Equation (4), with *c* = 0.96 ± 0.04.

**Figure 2 ijms-23-15392-f002:**
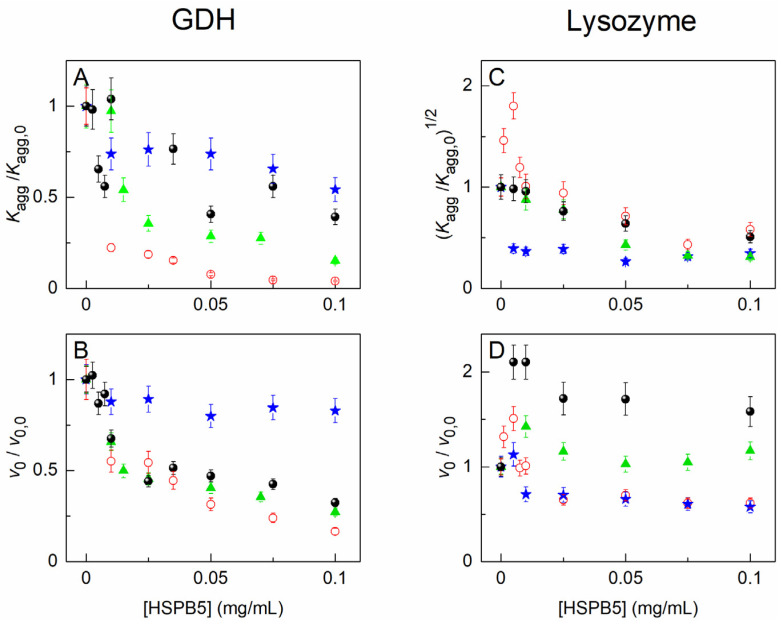
The effect of HSPB5 on the kinetics of thermal aggregation of GDH and DTT-induced aggregation of Lyz. (**A**,**B**) The dependences of *K*_agg_/*K*_agg,0_ and *v*_0_/*v*_0,0_ ratios, respectively, on HSPB5 concentration for aggregation of 0.2 mg/mL GDH at 50 °C (0.1 M Na-phosphate buffer, 10 mM NaCl, pH 7.6). (**C**,**D**) The dependences of (*K*_agg_/*K*_agg,0_)^1/2^ and *v*_0_/*v*_0,0_ ratios, respectively, on HSPB5 concentration for aggregation of 0.15 mg/mL Lyz at 37 °C (0.1 M Na-phosphate buffer, 10 mM NaCl, 20 mM DTT, pH 7.6). Open red circles correspond to the values obtained in the buffer; blue stars—in the presence of 25 mg/mL PEG; green triangles—in the presence of 1 M Bet; and filled black circles—in the presence of both 25 mg/mL PEG and 1 M Bet.

**Figure 3 ijms-23-15392-f003:**
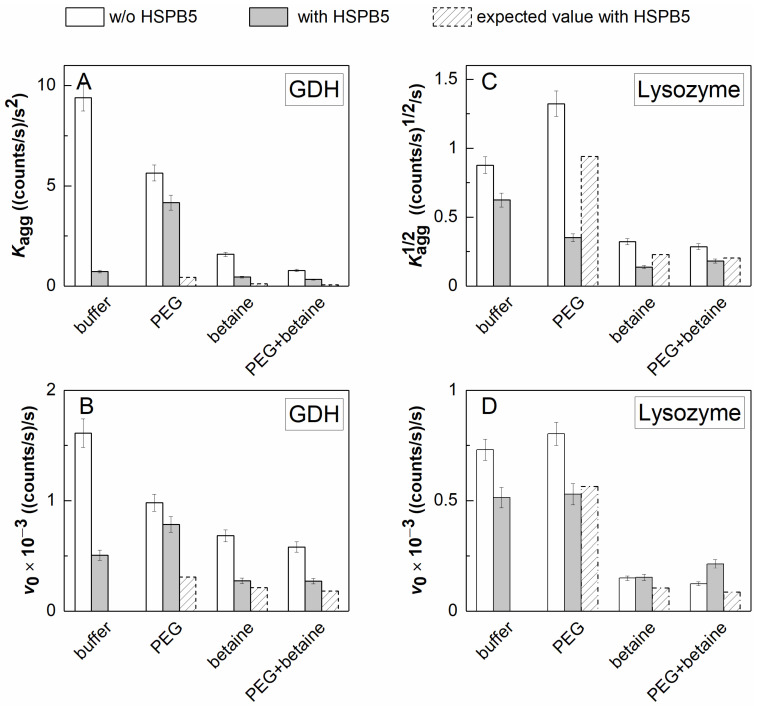
The comparison of HSPB5 effect on the kinetic parameters of the heat-induced aggregation of 0.2 mg/mL GDH at 50 °C and DTT-induced aggregation of 0.15 mg/mL Lyz at 37 °C in the absence and in the presence of 25 mg/mL PEG, 1 M Bet, and their mixture. Bar diagrams of (**A**) *K*_agg_ values for GDH, (**B**) *K*_agg_^1/2^ values for Lyz, and *v*_0_ values for (**C**) GDH and (**D**) Lyz. The hollow bars correspond to the experimental values for the samples without HSPB5; the gray bars—to the experimental values for the samples with 0.05 mg/mL HSPB5; and the shaded bars—to the expected values if the decrease under the action of HSPB5 is as big as in the buffer.

**Figure 4 ijms-23-15392-f004:**
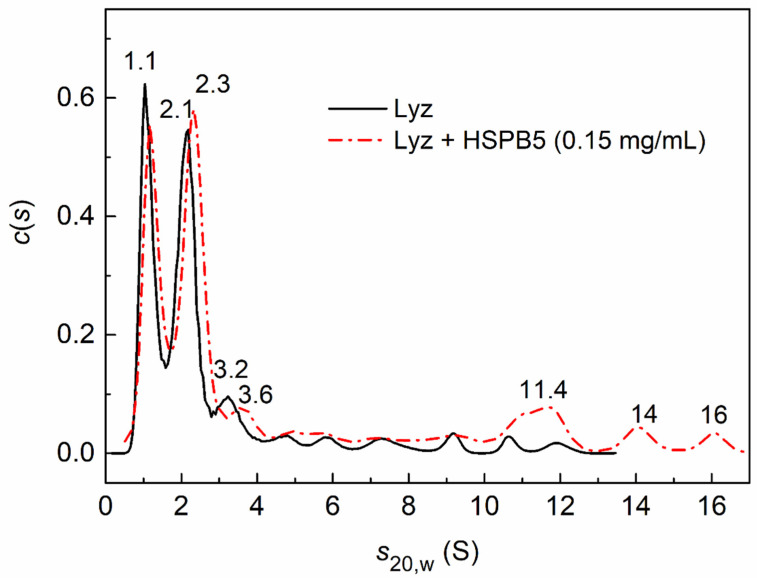
The sedimentation behavior of DTT-denatured Lyz (0.15 mg/mL) at 37 °C. The differential sedimentation coefficient distributions, *c*(*s*), for Lyz (black solid curve) and its mixture with HSPB5 (0.15 mg/mL, red dash-dotted curve) were obtained at 37 °C and corrected to standard conditions. Rotor speed was 60,000 rpm.

**Figure 5 ijms-23-15392-f005:**
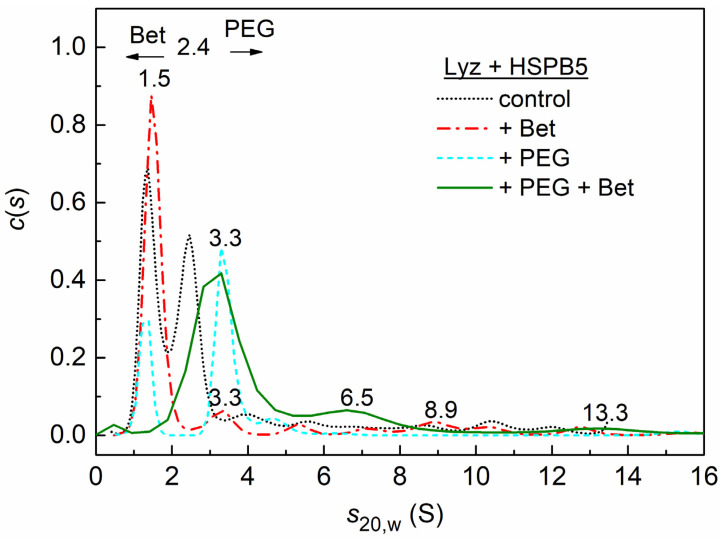
The sedimentation behavior of Lyz (0.15 mg/mL) in the presence of 0.05 mg/mL HSPB5. The differential sedimentation coefficient distributions, *c*(*s*), for the mixture of Lyz with HSPB5 (black dotted curve) with the addition of 25 mg/mL PEG (red dash-dotted curve), 1 M Bet (blue dashed curve), and 25 mg/mL PEG + 1 M Bet (green solid curve). All *c*(*s*) distributions were obtained at 37 °C and corrected to standard conditions. The rotor speed was 60,000 rpm for (Lyz + HSPB5) and (Lyz + HSPB5 + PEG) and 48,000 rpm for the other samples.

**Figure 6 ijms-23-15392-f006:**
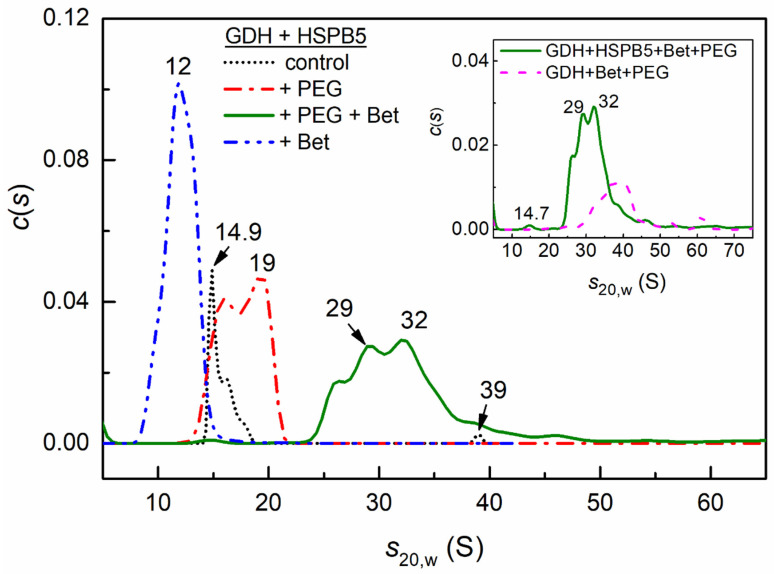
The sedimentation behavior of GDH (0.2 mg/mL, preheated 10 min at 50 °C) in the presence of 0.05 mg/mL HSPB5 at 25 °C. The differential sedimentation coefficient distributions, *c*(*s*), for the mixture of GDH and HSPB5 (black dotted curve) with the addition of 25 mg/mL PEG (red dash-dotted curve), 1 M Bet (blue dash-dot-dotted curve), and 25 mg/mL PEG + 1 M Bet (green solid curve). Inset shows *c*(*s*) distributions for GDH (0.2 mg/mL, preheated 10 min at 50 °C) in the presence of 25 mg/mL PEG + 1 M Bet (green solid curve) with the addition of 0.05 mg/mL HSPB5 (magenta dashed curve). All distributions were obtained at 25 °C and transformed to standard conditions. The rotor speed was 48,000 rpm.

**Table 1 ijms-23-15392-t001:** Estimation of the fraction of aggregated protein (γ_agg_) precipitated during the acceleration of the rotor in the AUC experiments at 25 °C. Rotor speed was 48,000 rpm. Before the experiment, the samples were heated at 50 °C for 10 min and then cooled in ice.

Sample	γ_agg_ (%)
GDH (0.2 mg/mL)	58
GDH + Bet (1 M)	0
GDH + Bet (1 M) + PEG (25 mg/mL)	50
GDH + Bet (1 M) + PEG (25 mg/mL) + HSPB5 (0.05 mg/mL)	0
GDH + Bet (1 M) + HSPB5 (0.05 mg/mL)	0
GDH + PEG (25 mg/mL) + HSPB5 (0.05 mg/mL)	0
GDH + HSPB5 (0.05 mg/mL)	57
GDH + PEG (25 mg/mL)	16

## Data Availability

Not applicable.

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
