# Peer review of "Effects of Molecular Crowding and Betaine on HSPB5 Interactions, with Target Proteins Differing in the Quaternary Structure and Aggregation Mechanism"

_ijms, 2022, doi:10.3390/ijms232315392_

Round 1
Reviewer 1 Report
Manuscript ID: ijms-2072776
Type of manuscript: Article
Title: Effect of Molecular Crowding and an Osmolyte on HspB5 Interaction with
Target Proteins Differing in the Quaternary Structure and Aggregation
Mechanism
In this research, the authors studied the effect of a crowding agent, PEG, and an osmolyte, betaine, on the antiaggregation activity of an sHSP, HSPB5. They showed that both agents affected the chaperone-like activity of HSPB5 and proposed that the change in oligomeric status of HSPB5 induced by the agents might contribute to their effects. Overall, this research is interesting to the readers in this field since it provide information regarding the action of chaperones in a complex system.
Specific comments:
1. Title, the title could be more specific to indicate the agent that studied in this research.
2. Line 53, molecular crowding can also be taken as one factor of the cellular environment.
3. Lines 218-223, the meanings of the labels have been indicated in the figure legend and it is unnecessary to state them again in the main text, which makes the sentences complicated and difficult to read.
4. Any explanation for the dissimilar effects of HSPB5 on the two substrates? Particularly, HSPB5 exhibit anti-chaperone effect at low concentrations on lysozyme aggregation, whereas no such effect on GDH aggregation. The dissimilarity or the substrate preference of HSPB5 might contribute to the dissimilar behaviors of the agents?
Reviewer 2 Report
The work of Borzova et al. presents interesting results on the impact of crowding agents, PEG and betaine, mimicking the cellular environment, on different stages of the aggregation of two model proteins, GDH and lysozyme, in the presence and absence of chaperone protein, HSPB5.
It is not clear why the authors, aiming to study the effect of the cellular environment on the aggregation process, decided to use different methods of denaturing model proteins. Why wasn't the same method chosen (lysozyme can be thermally denatured) for both proteins? In such a situation, it would be easier to generalize what impact crowding agents have on oligomeric and monomeric proteins, exemplified by GDH and lysozyme, respectively.
Besides, thermal denaturation does not introduce an additional element into the system under study, while a chemical agent, DTT, used to denature lysozyme, had to be removed before analytical centrifugation. Overnight dialysis and removal of the denaturant may have a significant effect on aggregation. Was it checked by a method other than AUC, if such an effect did not take place in the case of lysozyme?
What was the concentration of the stock solution of DTT? Did introducing DTT not significantly affect the final concentration of the remaining components of the tested systems, and if so, was their dilution properly addressed?
How could the authors explain the discrepancy between the ability of HSPB5 to inhibit aggregation and the similar amount of precipitating aggregates in the samples with and without the chaperone (58 and 57%, respectively, Table 1)?
One sentence seems not very accurate (line 418) because it may suggest that chaperone HSPB5 works more effectively in the presence of PEG than without it, which is true only in a narrow range of concentrations (below 0.025 mg/ml). For higher concentrations, HSPB5 works equally effectively in solutions with and without PEG.
